# Cytokine Profile in Development of Glioblastoma in Relation to Healthy Individuals

**DOI:** 10.3390/ijms242216206

**Published:** 2023-11-11

**Authors:** Pawel Jarmuzek, Piotr Defort, Marcin Kot, Edyta Wawrzyniak-Gramacka, Barbara Morawin, Agnieszka Zembron-Lacny

**Affiliations:** 1Department of Nervous System Diseases, Collegium Medicum, Neurosurgery Center University Hospital, University of Zielona Gora, 65-417 Zielona Gora, Poland; p.jarmuzek@cm.uz.zgora.pl (P.J.); m.kot@cm.uz.zgora.pl (M.K.); 2Department of Applied and Clinical Physiology, Collegium Medicum, University of Zielona Gora, 65-417 Zielona Gora, Poland; e.gramacka@cm.uz.zgora.pl (E.W.-G.); b.morawin@cm.uz.zgora.pl (B.M.); a.zembron-lacny@cm.uz.zgora.pl (A.Z.-L.)

**Keywords:** diagnostics, inflammation, neutrophil to lymphocyte ratio, prognosis, survival time

## Abstract

Cytokines play an essential role in the control of tumor cell development and multiplication. However, the available literature provides ambiguous data on the involvement of these proteins in the formation and progression of glioblastoma (GBM). This study was designed to evaluate the inflammatory profile and to investigate its potential for the identification of molecular signatures specific to GBM. Fifty patients aged 66.0 ± 10.56 years with newly diagnosed high-grade gliomas and 40 healthy individuals aged 71.7 ± 4.9 years were included in the study. White blood cells were found to fall within the referential ranges and were significantly higher in GBM than in healthy controls. Among immune cells, neutrophils showed the greatest changes, resulting in elevated neutrophil-to-lymphocyte ratio (NLR). The neutrophil count inversely correlated with survival time expressed by Spearman’s coefficient r_s_ = −0.359 (*p* = 0.010). The optimal threshold values corresponded to 2.630 × 103/µL for NLR (the area under the ROC curve AUC = 0.831, specificity 90%, sensitivity 76%, the relative risk RR = 7.875, the confidence intervals 95%CI 3.333–20.148). The most considerable changes were recorded in pro-inflammatory cytokines interleukin IL-1β, IL-6, and IL-8, which were approx. 1.5–2-fold higher, whereas tumor necrosis factor α (TNFα) and high mobility group B1 (HMGB1) were lower in GBM than healthy control (*p* < 0.001). The results of the ROC, AUC, and RR analysis of IL-1β, IL-6, IL-8, and IL-10 indicate their high diagnostics potential for clinical prognosis. The highest average RR was observed for IL-6 (RR = 2.923) and IL-8 (RR = 3.151), which means there is an approx. three-fold higher probability of GBM development after exceeding the cut-off values of 19.83 pg/mL for IL-6 and 10.86 pg/mL for IL-8. The high values of AUC obtained for the models NLR + IL-1β (AUC = 0.907), NLR + IL-6 (AUC = 0.908), NLR + IL-8 (AUC = 0.896), and NLR + IL-10 (AUC = 0.887) prove excellent discrimination of GBM patients from healthy individuals and may represent GBM-specific molecular signatures.

## 1. Introduction

Inflammation predisposes the development of cancer and supports all stages of tumorigenesis [1]. Tumor cells, as well as surrounding stromal and immune cells, are engaged in well-orchestrated mutual interactions to form an inflammatory tumor microenvironment. Neutrophils are the first cells to infiltrate under the direction of inflammatory mediators released at the inflammatory setting [2]. As the process of inflammation proceeds, lymphocytes and macrophages are stimulated and recruited to the site of inflammation. The immune cells within the tumor neutrophil microenvironment are highly plastic, and they continuously change their secretory phenotypic and functional characteristics [3]. Strong perturbation of tissue homeostasis leads to the recruitment of immune cells from bone marrow and secondary lymphoid cells, which results in an increase in counts of neutrophils, monocytes, and monocyte-derived cells in the peripheral blood [3]. Recently, some studies have reported combinations of immune cells, such as the neutrophil-to-lymphocyte ratio (NLR) and the lymphocyte-to-monocyte ratio (LMR), as effective prognostic indicators in patients with a variety of cancers [4,5,6,7,8,9,10]. When compared to traditional molecular prognostic markers, such as IDH1 mutation [11], the NLR can appropriately assess the prognosis in glioblastoma (GBM) patients in order to guide therapeutic decisions and patient management. A poorer prognosis was observed in the patients with NLR < 4.56 × 103/µL when compared with the patients with NLR < 4.56 × 103/µL [12]. Moreover, our last meta-analysis indicated the high diagnostic utility of peripheral immune-inflammatory markers for a poor prognosis in patients with GBM [13].

GBM is the most aggressive and also the most common brain tumor in adults, accounting for 80% of primary malignant brain cancers [14]. Standard treatment using maximal safe resection and chemoradiotherapy results in a median survival time of 14.6 months [15]. There is no standard second-line treatment and none that extends overall survival. The infiltrative pattern of growth and the inherent resistance of GBM to chemoradiotherapy leads to the disease recurring within 6 to 9 months of treatment. Survival of patients with recurrent GBM is generally less than 6 months [16,17]. Approximately 90% of GBM primarily occurs in older patients, whereas in younger patients, it usually develops from lower-grade glioma. Our retrospective study showed that the survival probability decreases considerably faster in older (63–90 years) than in young patients (23–63 years) with high-grade glioma [12]. As inflammation contributes to cancer initiation and progression, it can be hypothesized that age-related chronic low-grade inflammation contributes to the increase in cancer incidence and/or mortality observed during the aging process [3]. The inflammaging is revealed by the release of a large number of inflammatory mediators, such as pro-inflammatory cytokines IL-1β, IL-2, IL-6, IL-8, IL-12, IL-15, IL-18, IL-22, IL-23, tumor necrosis factor α (TNFα) and interferon γ (IFNγ), and anti-inflammatory cytokines IL-1Ra, IL-4, IL-10, and IL-13 [18,19]. The causal relationships between inflammaging and cancer are not straightforward, as these processes carry contrary features. Cellular senescence is an established tumor-suppressive mechanism that counteracts the proliferation of premalignant cells. However, much evidence has shown that senescent cells can also promote tumor progression in addition to other age-related pathologies via the senescence-associated secretory phenotype (SASP) [20]. Indeed, over the last few years, SASP has become one of the most important features of senescence and a key factor in our understanding of its complex and ambivalent relations with aging and cancer [21]. Several SASP components, including IL-1β, IL-6, IL-8, and TNFα, are all potential targets for pharmacological inhibition; some of them already exist in the clinical practice and may be exploited to mitigate the deleterious effects of SASP in the relevant clinical context [22]. Interestingly, these cytokines are the key inflammatory mediators that trigger the inflammatory cycle in GBM and also promote carcinogenesis by avoiding growth suppression and apoptosis, inducing angiogenesis and metastasis, and maintaining cancer cell stemness [2,23,24,25]. The concentration of cytokines in GBM cyst fluid highly correlates with white blood cell counts, suggesting an important interaction between tumor cells and the peripheral inflammatory status [24]. Cytokines released from the GBM or brain metastatic tissues and from immune cells in the tumor microenvironment can be identified in the circulation [26]. However, so far, two studies of peripheral cytokines have been performed in patients with glioblastoma, yet the obtained results have been inconclusive [26,27]. Cytokines are required to coordinate cancer-related processes and, therefore, represent an important subject of study for understanding GMB development and the potential identification of anti-inflammatory therapeutic targets. An immune-inflammatory profile should be characterized to be applied in clinical practice, and the profile's potential to identify GBM-specific molecular signatures should be investigated as a matter of urgency, not only to predict the outcomes in patients but also to recognize potential targets for future therapies.

## 2. Results

### 2.1. Study Population

Among the 50 study patients, 60% were females aged 67.4 ± 9.3 years with a median survival of 135 days (7–727 days), and 40% were males aged 63.1 ± 11.5 years with a median survival of 259 days (33–570 days). GBM was mostly located in the supratentorial region (frontal, temporal, and parietal lobes), with the highest incidence in the frontal lobe (32%). Ki-67 ≥ 30% recorded at 62%.

### 2.2. White Blood Cell Count-Derived Inflammation Indices

White blood cells were found to fall within the referential ranges and were significantly higher in GBM than HC. The counts of neutrophils and monocytes were significantly elevated, whereas lymphocytes were lowered in GBM. The neutrophil count inversely correlated with survival time r_s_ = −0.359 (*p* = 0.010). The platelet counts did not differ between groups. Among the investigated immune cells, neutrophils exceeded reference values and showed the greatest changes reflected by alternations in NRL and SII (Table 1). 

NRL and SII exceeded the reference values, whereas LMR and PLR were found to fall within the referential ranges proposed by Luo et al. [28]. NLR and SII inversely correlated with survival time r_s_ = −0.288 (*p* = 0.042) and r_s_ = −0.319 (*p* = 0.024), respectively. The results of the ROC analysis of NLR and SII were approx. 0.8, indicating a potential diagnostic value for clinical prognosis for patients with high-grade glioma (Figure 1A,D). The optimal threshold values corresponded to 2.630 × 10^3^/µL for NLR (AUC = 0.831, specificity 90%, sensitivity 76%, RR = 7.875, 95%CI 3.333–20.148), 3.22 × 10^3^/µL for LMR (AUC = 0.690, specificity 70%, sensitivity 65%, RR = 0.438, 95%CI 0.252–0.725), 143 × 10^3^/µL for PLR (AUC = 0.588, specificity 77.5%, sensitivity 54%, RR = 2.296, 95%CI 1.308–4.329), and 561 × 10^3^/µL for SII (AUC = 0.782, specificity 82.5%, sensitivity 70%, RR = 4.125, 95%CI 2.162–8.448) (Figure 1A–D).

### 2.3. Inflammatory Variables

The concentrations of IL-1β, IL-6, IL-8, IL-10, TNFα, and HMGB1 confirmed that inflammation was a critical component of glioblastoma progression (Table 2). 

The most considerable changes were recorded for pro-inflammatory cytokines IL-1β, IL-6, and IL-8, which were approx. 1.5–2-fold higher, whereas TNFα and HMGB1 levels were lower in GBM than HC (*p* < 0.001). Spearman’s test demonstrated a correlation between IL-1β, IL-6, IL-8, IL-10, and TNFα, while HMGB1 was inversely related to IFNγ in patients with GBM (Figure 2).

IL-1β highly correlated with IL-1Ra (r_s_ = 0.647, *p* < 0.0001), IL-6 (r_s_ = 0.352, *p* < 0.01), IL-8 (r_s_ = 0.409, *p* < 0.0001), and TNFα (r_s_ = 0.409, *p* < 0.001). Overall, our data confirms a cytokine milieu in glioma patients that favors the recruitment of neutrophils, thereby sustaining an immunosuppressive profile, which is associated with poor prognosis. The diagnostic capacity of inflammatory variables was determined by ROC curve analysis. The results of the ROC, AUC, and RR analysis of IL-1β, IL-6, IL-8, and IL-10 indicated a potential diagnostic value for clinical prognosis in patients with glioblastoma (RR > 2). The optimal threshold values (cut-off) corresponded to 0.469 pg/mL for IL-1β, 19.83 pg/mL for IL-6, 10.86 pg/mL for IL-8, and 41.22 pg/mL for IL-10 (Table 3). 

The highest mean RR was observed for IL-6 (RR = 2.923) and IL-8 (RR = 3.151), which means there is an approx. three-fold higher probability of GBM development once the cut-off values for these cytokines have been exceeded. Moreover, the highest specificity (80%) and sensitivity (80%) was observed for IL-6, which, in turn, indicates a low level of false positive results during diagnostic procedure using IL-6. Among the cytokines measured, IL-6 and IL-8 emerged as the best markers of inflammation-related GBM. The cytokines IL-1Ra, IL-13, and IFNγ showed poor diagnostic usefulness for prognosis in glioblastoma patients even though their concentrations changed significantly (Table 2 and Table 3). However, the predictive values increased when the models for pro- and anti-inflammatory cytokines included NLR or SII (Table 4 and Table 5). For the models NLR + cytokines models, AUC measurements were considered outstanding discrimination for NLR + IL-1β (AUC = 0.907), NLR + IL-6 (AUC = 0.908), and NLR + IL-8 (AUC = 0.896), and in the other models the values of AUC > 0.8 provided excellent discrimination [29]. The classifier accuracy for the models NLR + cytokines models was >80% (Table 4) and was higher than the models including SII + cytokines >70% (Table 5).

A high value of a classifier renders it a reliable index to distinguish a healthy individual from a GBM patient based on the assessment of inflammatory markers. Therefore, it seems reasonable to assess the combinations of immune cells, such as the neutrophil-to-lymphocyte ratio and inflammatory mediators, together.

## 3. Discussion

Chronic inflammation allows developing tumors to acquire all their characteristic abilities, including the escape from immunosurveillance. Cellular and molecular inflammatory mediators, such as growth factors and inflammatory cytokines produced by tumor cells and infiltrating immune cells, constitute the tumor microenvironment, where tumor cells constantly grow and interact. The heterogeneity and plasticity of tumor-associated neutrophils (TANs) render them crucial in the tumor microenvironment interplay. Increasing evidence suggests a dual modulatory role of neutrophils in tumor behavior and highlights the need for a reassessment of neutrophil functions in cancer initiation and progression [30]. Tumor-derived cytokines induce the presence in the blood of immature neutrophils with immunosuppressive properties and neutrophils with an ‘aged’ phenotype that are experienced cells with an increased ability to react to inflammatory stimuli, which can thus play an anti-tumoral role [31,32]. Moreover, neutrophils in the tumor tissue occur in different polarized states, i.e., N1 anti-tumoral phenotype and N2 pro-tumoral phenotype, analogous to helper T lymphocyte and monocyte polarization [33]. Despite detected functional differences, no definitive surface markers have been identified to differentiate N1 and N2 TANs [34]. Clinical studies have shown that most glioma patients experience strong neutrophilia, and that preoperative neutrophil count is correlated with GBM grade, but the mechanism of neutrophil recruitment and their role in tumor growth is yet to be defined [35]. We observed that among immune cells, neutrophils showed the most considerable changes, especially in patients with Grade 3 and Grade 4 tumors, whereas lymphocytes, monocytes, and platelets did not exhibit significant changes compared to reference levels or Grade 1 group [12]. The present study showed that the neutrophil count exceeded the reference values and was elevated in patients with GBM compared to the control group. Neutrophil counts inversely correlated with survival time r_s_ = −0.359 (*p* = 0.010), which clearly shows that patients with neutrophilia run a higher risk of mortality. Lymphocyte counts in our study were significantly lower, which was reflected in a high rate of NLR exceeding the reference values according to Luo et al. [28] and was four-fold higher in patients with GBM than in the control group. According to Massara et al. [33], NLR higher than 4 was associated with poor prognosis when measured before treatments and with increased TANs infiltration; however, the underlying mechanism remains unknown [36,37,38,39]. The NLR value below 4 was reported to predict better outcomes but only in GBM expressing the wild-type gene IDH1, one of the genes that is most frequently mutated in malignant gliomas [40]. Furthermore, the retrospective study by Mason et al. [41] recorded the link between lower NLR values and longer overall survival during focal radiotherapy and concomitant temozolomide treatment. In our previous analyses of inflammatory indices, NLR was found to exert the most substantial impact on survival time (HR 1.56; 95%CI 1.145–2.127; *p* = 0.005), and this index was superior to LMR, PLR, and SII as prognostic factors. With regard to age, the survival probability decreased by 50% in patients aged ≥63 years and NLR ≥ 4.56 × 103/µL [12]. 

Numerous reports showed how different age-related pathologies, including cancer, revealed a common inflammatory status. In fact, inflammaging is characterized by the establishment of a systemic proinflammatory state with an increased level of well-known pro-tumorigenic cytokines, such as IL-1β, IL-6, and TNFα [19,42]. In the elderly with high-grade inflammation (CRP ≥ 3 mg/L), elevated levels of pro-inflammatory cytokines IL-1β, IL-6, and TNFα were recorded when compared to older adults with low-grade inflammation (CRP < 3 mg/L) [19]. The cytokine profile of our GBM patients was found to be extremely different from that of the healthy controls representing inflammaging. The cytokines IL-1β, IL-1Ra, IL-6, IL-8, IL-10, and IL-13 were higher, whereas TNFα, IFNγ, and HMGB-1 were lower in GBM compared to control older adults. The pro-inflammatory cytokines IL-1β, IL-6, and IL-8 were two-fold elevated in contrast to TNFα, which appears to be a cytokine discriminating the inflammatory response in aging and cancer. The comparison of patients with GBM according to the cut-off value of 63 years established in our previous studies showed significantly lower levels of TNFα (51.35 ± 26.97 pg/mL) in patients aged ≤63 years than in patients aged >63 years (62.53 ± 16.76 pg/mL) [12]. Glioma-derived factors, such as IL-1β, IL-6, IL-8, and TNFα, are crucial inflammatory mediators that trigger the inflammatory cycle in GBM and also promote carcinogenesis by avoiding growth suppression and apoptosis, inducing angiogenesis and metastasis and maintaining cancer cell stemness [23,43]. Furthermore, glioma-derived cytokines IL-1β, IL-6, and TNFα were observed to extend neutrophils lifespan from 7 h in normal conditions to 17 h in cancer, which in turn increases the number of neutrophils in peripheral blood [44].

Cytokines, such as IL-1β and TNFα, are primarily pro-inflammatory and play an important role in inflammation-driven tumor growth and progression and are found to be upregulated after radiotherapy in patients with GBM. IL-1β coordinates the progression of neuroinflammation by upregulating the expression of other pro-inflammatory cytokines, whereas TNFα induces the expression of vascular endothelial growth factor in gliomas, leading to increased angiogenesis seen in GBM [45]. IL-1β binds to IL-1 receptor (IL-1R) and switches on the NF-kB pathway, leading to persistent stimulation of pro-inflammatory genes [46]. This could explain the simultaneous increase in IL-1Ra, IL-6, and IL-8 in patients with GBM compared with the control group; IL-1β highly correlated with IL-1Ra, IL-6, and IL-8 (*p* < 0.001). IL-1β and IL-1Ra are mainly expressed by cells of the monocyte/macrophage lineage and neutrophils. The constitutive expression of both IL-1β and IL-1Ra has been reported in human glioma cells [47]. The model for IL-1β, including NLR (AUC = 0.907, Cut-off 0.473, Classifier accuracy 85.6%), showed a high diagnostic potential in glioblastoma and confirmed an implication of IL-1β in neutrophils recruitment and tumor promotion. IL-1β is a well-known cytokine that upregulates mRNA expression of pro-inflammatory cyclooxygenase 2 (COX-2), which, together with prostaglandin PGE2, interacts with PGE2 receptors, and thereby enhances glioma aggressiveness by maintaining glioma cell stemness and the inflammatory microenvironment [48,49,50]. TNFα acts in a similar manner to IL-1β and plays various roles in the inflammatory response, including the activation of inflammatory cytokines coded by the NF-κB signal pathway, gene expression of prostaglandin synthesis pathway enzymes, adhesion molecules and induction of nitric oxide synthase, leading to the activation of endothelium, neutrophils, and lymphocytes [51]. TNFα is related to all steps in tumorigenesis, including cellular transformation, promotion, survival, proliferation, invasion, angiogenesis, and metastasis. Therefore, it was surprising that circulating TNFα concentration was reduced in the GBM group compared to healthy controls (*p* < 0.001), especially when we take into account the meta-analysis by Feng et al. [52], which showed that elevated circulating TNFα level was associated with an increased glioma risk. On the other hand, the same study showed that in all the cytokines tested, it was the circulating IL-6 level (HR 1.10, 95%CI 1.05–1.16, *p* < 0.001) that was the most significantly correlated with poor overall survival in glioma patients. Rubenich et al. [25] demonstrated that glioma-neutrophil cultures initially produced substantial amounts of TNFα, which then kept proportionally decreasing over the hours while the IL-6 level increased 10-fold from the baseline. 

IL-6 is involved in many physiological and pathological processes, including inflammation, bone metabolism, C-reactive protein synthesis, hematopoiesis, leucocyte infiltration, and maturation, as well as affecting endothelial cell properties. IL-6 was also found to induce cachexia in cancer patients by altering the metabolism of lipids and proteins [17,53]. IL-6 production is characteristic of glioblastoma cells, and its upregulation is associated with activation and repolarization of tumor-associated neutrophils. Some studies have also shown a relationship between the proliferative antigen Ki-67 and IL-6 in the surgical samples [54]. High IL-6 gene expression in glioblastoma is associated with poor survival according to the datasets derived from The Cancer Genome Atlas (TCGA) and the Repository of Molecular Brain Neoplasia Data [55]. The present work aimed at studying the prognostic value of changes in circulating IL-6 levels for glioblastoma by comparing it to other cytokines in GBM patients and healthy controls. The highest average RR was observed for IL-6 (RR = 2.923) and IL-8 (RR = 3.151), which means approx. There is a three-fold higher probability of GBM development after exceeding the cut-off values for these cytokines. Moreover, IL-6 was observed to have the highest specificity (80%) and sensitivity (80%), which, in turn, indicates a low level of false positive results during diagnostic procedures using IL-6. Among the cytokines measured, IL-6 emerged as the best marker of inflammation-related glioblastoma. The model for IL-6, including NLR, showed the highest diagnostic potential (AUC = 0.908, Cut-off 0.544, Classifier accuracy 87.8%). 

Immunohistochemical studies have exposed that about 65% of primary and secondary GBM samples are directly correlated with IL-8 levels [56]. IL-8 induces angiogenesis and directs migration of endothelial cells, further stimulating the production of proteolytic enzymes-matrix metalloproteinases [57]. In this study, the levels of circulating IL-8 were found to be higher in GBM patients than in healthy controls (*p* < 0.001). Many studies reported IL-8 upregulation in gliomas and its involvement in disease promotion [56,58,59]. Rubenich et al. [25] showed that glioma–neutrophil cultures produced large amounts of IL-8 and TNFα and from low to zero levels of IL-1β, IL-6, and IL-10 in the first 24 h. Then, IL-8 and IL-10 increased, while the production of TNFα was observed to decrease after 72 h. However, after 120 h, this pattern shifted, and a rise in IL-1β and IL-6 release was recorded, while IL-8 declined and IL-10 and TNFα were barely detectable. All control neutrophils, regardless of the culture duration, showed very low or no cytokine production. Overall, the data provided by Rubenich et al. [25] proved that the neutrophil–glioblastoma crosstalk was accountable for the diversity of the activities of the cytokines described as drivers of tumor progression. 

IL-10 and IL-13 are classified as anti-inflammatory cytokines. Early evidence for the expression of IL-10 in glioma was reported by Huettner et al. [60], who demonstrated elevated levels of IL-10 mRNA in 87% of high-grade gliomas. In our study, IL-10 was elevated in GBM patients, as was observed in several other studies that revealed its inhibitory effect on the antitumor response while promoting the proliferation of tumor cells [60,61,62]. Interleukin IL-13 is expressed in glioblastoma; it binds to two receptors, IL-13Rα1 and IL-13Rα2, and mediates a variety of different effects on various cell types, including B cells, neutrophils, monocytes, natural killer cells, endothelial cells, and fibroblasts [63,64]. The overexpression of IL-13Rα2 was observed in approx. 76% of GBM, but it was not detected in normal brain tissue, thereby making it a highly selective immunotherapy target [65]. There is limited evidence linking IL-10 and IL-13 expression with GBM patient survival. In our GBM patients, the elevated IL-10 levels *(p* < 0.01) correlated with IL-13 (r_s_ = 0.416, *p* < 0.01), which was associated with increased mortality according to most studies on elderly populations [66]. At first glance, the strong increase in IL-10 looks paradoxical in the balance of pro- and anti-inflammatory cytokines; however, the finding confirms the previous study by Kumar et al. [67], who reported a significant increase in serum IL-10 levels in patients with anaplastic astrocytoma and glioblastoma. Bender et al. [26] used a multiplex immunoassay platform and demonstrated at least double IL-10 levels in GBM patients compared to healthy controls, whereas the levels of other cytokines, i.e., IL-6, IL-8, and TNFα, did not differ between groups. Overall, several studies have confirmed the protumor properties of IL-10 [68]. Elevated levels of IL-10 and NLR are associated with increased tumor growth with poor prognosis and drug resistance [25]. In our NLR + IL-10 model, the high value of AUC = 0.887 proved to offer excellent discrimination of GBM patients from healthy individuals based on the assessment of both markers. On the other hand, IL-10 inhibits tumorigenesis via downregulation of IL-1β, IL-6, IL-8, and TNFα thus playing an important role in coordinating the inflammatory response involving the activation of neutrophils, monocytes, natural killer cells, and T and B cells and in their recruitment to the sites of inflammation. Despite reducing tumor-promoting inflammation, IL-10 may play a role in the rejuvenation of exhausted tumor-resident T cells [69]. The obtained cut-off value (41.22 pg/mL) could serve as a biomarker indicative of GBM development and its prognosis (AUC = 0.683, specificity 67.5%, sensitivity 72.0%, RR = 2.482, 95%CI 1.519–4.222). Similarly, IL-13 largely inhibits tumor cell growth, but recent studies revealed that it could promote the survival of certain tumors through suppression of immunosurveillance [70]. These apparently contradictory results may be explained by diverse IL-13 activity in different tumors and the fact that earlier studies investigated tumor cell lines rather than primary tumor cells. In our study, IL-13 levels tended to increase in GBM patients compared to healthy individuals (*p* > 0.05). Although IL-13 hardly emerged as a good biomarker in glioblastoma (AUC = 0.536, specificity 85%, sensitivity 38%, RR = 0.459, 95%CI 0.213–0.888), its diagnostics usefulness increased in NLR + IL-13 model to AUC = 0.830 and classifier accuracy >80%.

IFNγ is a cytokine that consistently orchestrates both pro-tumorigenic and antitumor immunity. IFNγ acts together with granzyme B and perforin to initiate apoptosis in tumor cells but also enables the synthesis of immune checkpoint inhibitory molecules, such as PD-L1, thus stimulating other immune-suppressive mechanisms [71]. IFNγ has also been shown to downregulate other IL-1β-mediated effects, such as the expression of IL-6, IL-8, and IL-10; however, the mechanism by which IFNγ mediates these inhibitory effects has not yet been determined [72]. In our study, no differences in IFNγ levels were observed between groups. Actually, its level even tended to be lower in GBM patients (Table 2), and this cytokine provided a low diagnostics value for clinical prognosis in GBM patients (AUC = 0.523, specificity 70%, sensitivity 48%, RR = 0.643, 95%CI 0.370–1.056). IFNγ was inversely correlated with HMGB1, but in patients with GBM, the correlation reached a moderate level (r_s_ = −0.294, *p* < 0.05). Nijaguna et al. [27] demonstrated high expression of IFNγ and 14 other cytokines, including IL-6, IL-10, and TNFα. Bender et al. [26] used a multiplex immunoassay platform and demonstrated a two-fold increase in IFNγ level in GBM patients compared to healthy controls, whereas other cytokines, i.e., IL-6, IL-8, and TNFα, showed no differences between groups [26]. The differences between our findings and those from previous cytokine profiling studies [26,27] may have resulted from sampling differences, i.e., plasma vs. serum through the course of a patient’s disease, and various measurement methods, i.e., multiplex vs. singleplex immunoassay. Undoubtedly, multiplex proteomic immunoassay procedures are the future of diagnostics testing but require extensive validation before being acceptable for clinical use. Regardless of different data on IFNγ concentration changes, it could be used in combination therapy in cancer, including glioblastoma [73,74,75]. 

HMGB1 is a pro-inflammatory cytokine that has been extensively studied for the past years as a biomarker and a novel target for cancer therapies [76]. Accumulating evidence has demonstrated that hypoxia, reactive oxygen, nitrogen species, hyperglycemia, cytokines TNFα, and INFγ induce tumor cells to actively secrete HMGB1 into the extracellular matrix [77]. Subsequently, extracellular HMGB1 (in its reduced form) functions as a paracrine and/or autocrine factor to activate signaling cascades by binding to its receptors, such as the receptor for advanced glycation end-products (RAGE) and toll-like receptors (TLR) [78]. The RAGE-HMGB1 axis is a major aspect of immune signaling in pathogenic conditions such as glioma and inflammatory diseases [79]. By binding to RAGE and TLR, HMGB1 can mediate multiple inflammatory pathways, and it induces secretion of different pro-inflammatory cytokines like IL-1β, TNFα, IL-6, and IL-8 in the culture medium of human monocytes and neutrophils [80]. HMGB1 is also a DNA-binding nuclear protein that activates genes for inflammatory cytokines, such as IL-1β, TNFα, IL-6, and IFNγ [76,79]. In our study, changes in circulating HMGB1 were comparable to TNFα and INFγ, i.e., HMGB1 decreased in GBM patients compared to healthy control (*p* < 0.001). There are no available data on extracellular or circulating HMGB1 levels from human studies [81]. HMGB1 has been shown to be highly expressed in human glioma cells and to be associated with poor prognosis [82,83]. The level of HMGB1 expression in gliomas is three-fold higher than in normal brains, and during inflammation, HMGB1 may be released extracellularly from neurons, glial, or endothelial cells [81,83]. On balance, the extracellular release of HMGB1 may play a very important role in triggering initial inflammatory responses by stimulating multiple receptors, leading to blood–brain barrier disruption. The univariate logistic model demonstrated the average diagnostics value of circulating HMGB1 (AUC = 0.708, specificity 75%, sensitivity 66%, RR = 0.364, 95%CI 0.199–0.628) compared to other inflammatory variables. The comparison of our results with the outcomes obtained by other authors allowed us to conclude that HMGB1 measurement may be useful in the assessment of clinical management of GBM patients on condition that the management includes chemo- and radiotherapy and the ensuing inflammatory conditions and necrosis [81,84,85]. Alas, patients undergoing chemotherapy or radiotherapy were excluded from our observations. Nevertheless, our understanding of the interplay between peripheral inflammation and HMGB1 in the progression and prognosis of glioblastoma is essential and requires further research.

## 4. Materials and Methods

### 4.1. Study Population

The study was carried out on 50 patients aged 66.0 ± 10.56 years (females *n* = 30, males *n* = 20) with newly diagnosed glioblastoma who had undergone an operation in Neurosurgery Centre University Hospital in Zielona Gora between November 2015 and May 2021 (Table 6). There were no demonstrated extracranial metastases in newly diagnosed glioblastoma during the follow-up period.

The pathological diagnosis was based on the classification of CNS tumors [86]. The overall survival was defined as the time between the diagnosis and death. For the patients who had not died prior to the last follow-up, the overall survival was censored at the date of the last follow-up. All patients underwent a craniotomy on GBM with either total or subtotal resection. The following exclusion criteria were used: biopsy only, age below 18 years, no definite diagnosis, incomplete baseline clinical data, adjuvant therapy like chemotherapy or radiotherapy received before the operation, malnutrition, and perioperative mortality. Importantly, every patient diagnosed with a primary brain tumor and registered in our study had a very recent diagnosis with no prior specific treatment, including glucocorticoids. The GBM group was compared to the healthy control (HC) of 40 individuals aged 71.7 ± 4.9 years (males *n* = 18 and females *n* = 22) recruited from the University of the Third Age in Zielona Gora (Poland), which is an organization associating the adults over 60 years of age to stay active by participating in many educational and sports programs. The current health status of the control group was evaluated on the basis of medical records at a routine follow-up visit to a primary care physician. On the basis of the medical interview, the following exclusion criteria were applied: uncontrolled hypertension and/or diabetes, oncologic diseases and neurodegenerative diseases, and acute infectious and autoimmunological diseases. The study protocol was approved by the Bioethics Commission at the University of Zielona Gora, Poland (No. UZ19/2021, No. UZ16/2022), in accordance with the Helsinki Declaration.

### 4.2. Clinical Assessment

Medical records were reviewed, and clinical data were collected, such as gender, age at operation, the location and hemisphere of tumors, and pathological diagnoses. Ki-67 proliferation index was expressed as the percentage of cells with Ki-67-positive immunostained nuclei using the Ventana BenchMark GX (Ventana Medical Systems Inc., Tucson, AZ, USA). The expression of Ki-67 was categorized into two groups: low and intermediate (Ki-67 < 30%) and high (Ki-67 ≥ 30%), according to Chen et al. [87]. C-reactive protein was measured using a high-sensitivity commercial kit from DRG International (Springfield Township, Cincinnati, OH, USA) with a detection limit of 0.001 mg/L. The date on postoperative adjuvant therapies and survival time were collected through documentation analysis.

### 4.3. Blood Sampling

Fasting blood samples were collected from the median cubital vein in the morning between 07:00 and 09:00 using the S-Monovette system (Sarstedt AG & Co. KG, Nümbrecht, Germany). The whole blood samples were placed into tubes with anticoagulant EDTA-K2 and were immediately analyzed. For the other biochemical measures, blood samples were centrifuged at 3000 rpm for 10 min, and aliquots of serum were stored at −80 °C in a Clinic BioBank until analysis.

### 4.4. White Blood Cell Count-Derived Inflammation Indices

Hematological variables, including total white blood cell count (WBC), platelet count, and differential WBC, were determined by Sysmex XN-1000 (Sysmex Europe GmbH, Norderstedt, Germany). The neutrophil-to-lymphocyte ratio (NLR × 10^3^/µL), the lymphocyte-to-monocyte ratio (LMR × 10^3^/µL), the platelets-to-lymphocytes (PLR × 10^3^/µL), and the systemic immune inflammation index (SII × 10^3^/µL = (platelets × neutrophils)/lymphocytes)) were calculated and compared to reference values according to Luo et al. [28].

### 4.5. Inflammatory Variables

Cytokines IL-1β, IL-1Ra, IL-6, IL-8, IL-10, IL-13, IFNγ, TNFα, and high mobility group box 1 (HMGB1) concentrations were determined using ELISA kits from SunRed Biotechnology Company (Shanghai, China) with detection limits of 28.384 pg/L, 28.125 ng/L, 1.867 pg/mL, 1.869 ng/L, 1.142 pg/mL, 5.436 pg/mL, 1.706 ng/L, 2.782 ng/L, and 0.526 ng/mL, respectively. The average intra-assay coefficients of variation (intra-assay CV) for the used enzyme immunoassay tests (ELISA) were <8%. All samples were analyzed in duplicate in a single assay to avoid inter-assay variability.

### 4.6. Statistical Analysis

Statistical analyses were performed using R 4.2.1 software [88]. The variables were reported as mean values ± standard deviation (SD) and median (Me). The assumptions for the use of parametric or nonparametric tests were checked using the Shapiro–Wilk and Levene’s tests to assess the normality of the distributions and the homogeneity of variances, respectively. The significant differences in mean values between the groups were evaluated by one-way ANOVA. If the normality and homogeneity assumptions were violated, the Mann–Whitney nonparametric test was used. Spearman’s rank correlation (r_s_ Spearman’s rank correlation coefficient) was used to investigate the relationships between immune-inflammatory markers. The predictive value of inflammatory variables was evaluated using the receiver operating characteristic curve (ROC). Area under the ROC Curve (AUC) was used to provide an aggregate measure of performance across all possible classification thresholds. Both univariate and multivariate logistic regression models were used. The optimal threshold value for clinical stratification (cut-off value) was obtained by calculating the Youden index. Relative risk (RR) was performed for univariate analyses. Statistical significance was set at *p* < 0.05.

## 5. Conclusions

Glioblastoma promotes an immune-inflammatory response that could shift the tumor microenvironment into a pro-tumorigenic milieu. Acute and temporary inflammation may inhibit tumor growth by upregulating inflammatory mediators, such as interleukins IL-1β, IL-6, IL-8, TNFα, IFNγ, and HMGB1 as pro-inflammatory cytokines, and IL-1Ra, IL-10, and IL-13 as anti-inflammatory cytokines that are part of the initial inflammatory cascade and recruit other downstream targets to enhance antitumor responses. However, if inflammation becomes a chronic state, the same inflammatory processes can exhaust the immune system's ability to fight against glioblastoma and induce the release of immature neutrophils with immunosuppressive properties. The cytokine profile in our research appeared to vary independently of the primary pro- or anti-inflammatory activity, i.e., IL-1β, IL-6, IL-8, and IL-10 levels were elevated, whereas TNFα and HMGB1 were decreased in GBM patients compared to healthy control. We assume that interactions between neutrophils and glioma cells are responsible for the diverse activities of the cytokines described as drivers of tumor progression. The following cytokines, IL-1β, IL-6, IL-8, and IL-10, unlike IL-1Ra, IL-13, and IFNγ, demonstrated high diagnostic potential for clinical prognosis for GBM patients, especially in models with neutrophil-to-lymphocyte ratio. Although inflammation is highly relevant in our understanding of the pathology of GBM, the immune-inflammatory profile has not been sufficiently defined and still requires further research.

## Figures and Tables

**Figure 1 ijms-24-16206-f001:**
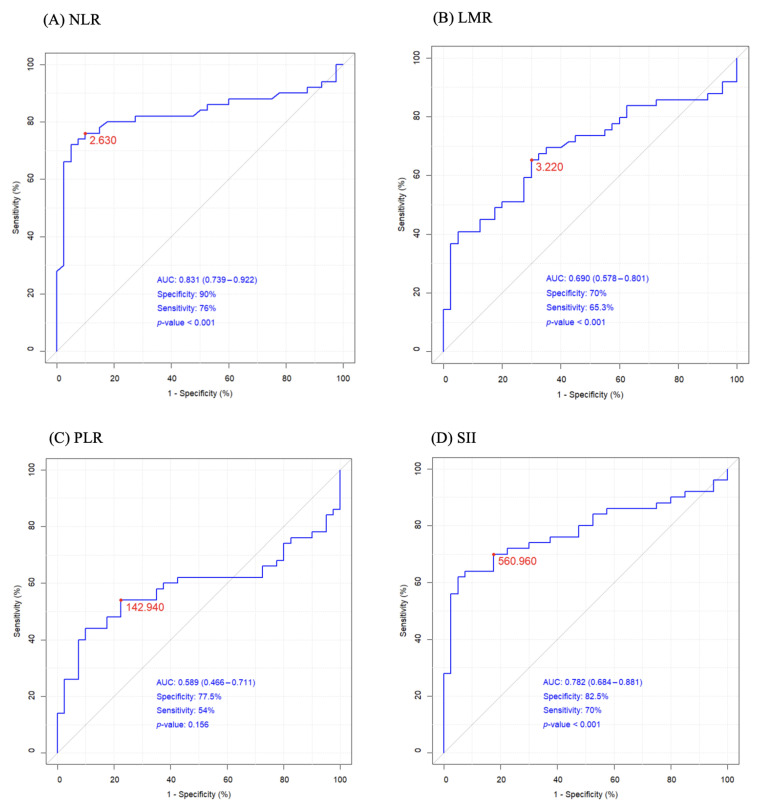
The ROC curves of (**A**) NLR neutrophil/lymphocyte ratio, (**B**) LMR lymphocyte/monocyte ratio, (**C**) PLR platelets/lymphocytes ratio, (**D**) SII systemic immune inflammation index.

**Figure 2 ijms-24-16206-f002:**
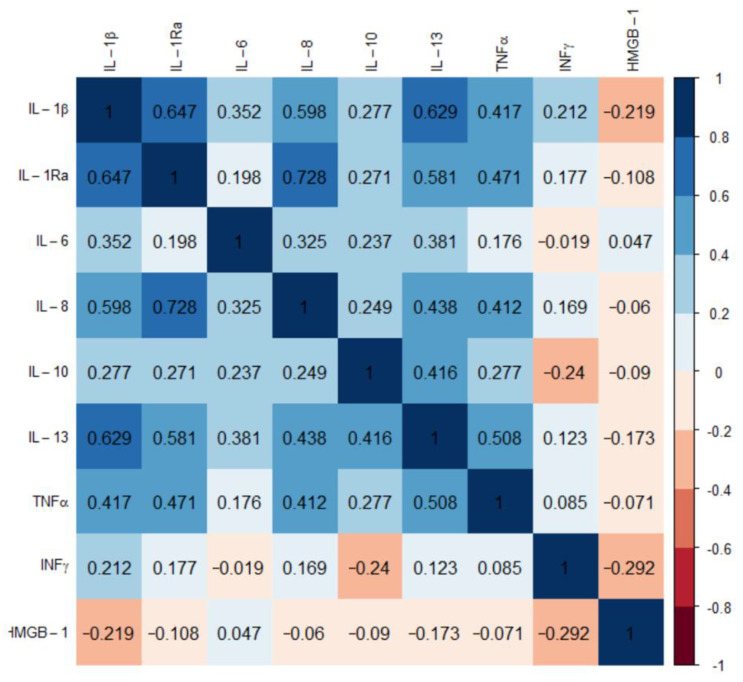
Heat map with correlation analysis between levels of cytokines in glioblastoma (*n* = 50); Spearman’s rank correlation coefficient.

**Table 1 ijms-24-16206-t001:** White blood cell count-derived inflammation indices.

Index	Reference Values	GBM *n* = 50 Mean ± SD (Me)	HC *n* = 40 Mean ± SD (Me)	GBM vs. HC *p* Level
WBC (10^3^/µL)	4.0–10.2	8.41 ± 2.92 (7.86)	6.03 ± 1.54 (6.06)	<0.001
Neutrophils (10^3^/µL)	2.0–6.9	9.95 ± 8.46 (7.52)	3.37 ± 1.31 (3.04)	<0.001
Lymphocytes (10^3^/µL)	0.6–3.4	1.75 ± 1.27 (1.42)	1.90 ± 0.58 (1.82)	0.022
Monocytes (10^3^/µL)	0.00–0.90	0.69 ± 0.73 (0.52)	0.50 ± 0.12 (0.49)	0.555
Platelets (10^3^/µL)	140–420	216 ± 109 (206)	219 ± 58 (209)	0.873
NLR (10^3^/µL)	0.87–4.15	8.08 ± 7.26 (5.53)	1.98 ± 1.53 (1.74)	<0.001
LMR (10^3^/µL)	2.45–8.77	4.21 ± 5.90 (2.82)	3.97 ± 1.27 (3.73)	<0.001
PLR (10^3^/µL)	47–198	173 ± 120 (151)	123 ± 46 (112)	0.143
SII (10^3^/µL)	142–808	1785 ± 1769 (1214)	442 ± 375 (343)	<0.001

Abbreviations: GBM glioblastoma, HC healthy control, WBC white blood cells, NLR neutrophil/lymphocyte ratio, LMR lymphocyte/monocyte ratio, PLR platelets/lymphocytes ratio, SII systemic immune inflammation index.

**Table 2 ijms-24-16206-t002:** Inflammatory variables.

Variable	GBM *n* = 50 Mean ± SD (Me)	HC *n* = 40 Mean ± SD (Me)	GBM vs. HC *p* Level
IL-1β pg/mL	0.641 ± 0.232 (0.616)	0.459 ± 0.183 (0.433)	<0.001
IL-1Ra pg/mL	69.83 ± 32.86 (62.83)	63.42 ± 20.76 (59.92)	0.581
IL-6 pg/mL	22.51 ± 8.30 (22.51)	16.25 ± 5.66 (12.27)	<0.001
IL-8 pg/mL	17.03 ± 9.35 (13.57)	10.32 ± 2.62 (9.62)	<0.001
IL-10 pg/mL	49.67 ± 18.39 (49.06)	39.35 ± 9.71 (38.41)	0.003
IL-13 pg/mL	5.83 ± 2.66 (5.37)	5.78 ± 1.67 (5.39)	0.561
TNFα pg/mL	57.84 ± 22.56 (54.80)	71.20 ± 18.04 (73.22)	<0.001
IFNγ pg/mL	161 ± 101 (174)	174 ± 67 (186)	0.712
HMGB1 ng/mL	44.47 ± 33.76 (30.83)	51.03 ± 31.14 (40.05)	<0.001

Abbreviations: GBM glioblastoma, HC healthy control, IL-1β interleukin 1β, IL-1Ra interleukin-1 receptor antagonist, IL-6 interleukin 6, IL-8 interleukin 8, IL-10 interleukin 10, IL-13 interleukin 13, TNFα tumor necrosis factor α, IFNγ interferon γ, HMGB1 high mobility group box 1.

**Table 3 ijms-24-16206-t003:** The statistical characteristics of the ROC curve for the univariate logistic model for inflammatory variables and the relative risk (RR) and its confidence intervals (95% CI) for indications determined by the cut-off values calculated on the ROC curves.

Variable	AUC	Cut-Off Value	Specificity (%)	Sensitivity (%)	RR	95% CI	*p* Level
IL-1β pg/mL	0.720	0.469	55.0	78.0	2.111	1.346–3.342	<0.001
IL-1Ra pg/mL	0.525	61.31	57.5	56.0	1.353	0.851–2.191	0.685
IL-6 pg/mL	0.725	19.83	80.0	80.0	2.923	1.605–5.724	<0.001
IL-8 pg/mL	0.735	10.86	72.5	76.0	3.151	1.867–5.588	<0.001
IL-10 pg/mL	0.683	41.22	67.5	72.0	2.482	1.519–4.222	<0.05
IL-13 pg/mL	0.536	4.45	85.0	38.0	0.459	0.213–0.880	0.560
TNFα pg/mL	0.720	65.72	67.5	74.0	0.385	0.227–0.629	<0.001
IFNγ pg/mL	0.523	159.24	70.0	48.0	0.643	0.370–1.056	0.712
HMGB1 ng/mL	0.708	34.52	75.0	66.0	0.364	0.199–0.628	<0.001

Abbreviations: AUC area under the ROC curve, IL-1β interleukin 1β, IL-1Ra interleukin-1 receptor antagonist, IL-6 interleukin 6, IL-8 interleukin 8, IL-10 interleukin 10, IL-13 interleukin 13, TNFα tumor necrosis factor α, IFNγ interferon γ, HMGB1 high mobility group box 1.

**Table 4 ijms-24-16206-t004:** The characteristics of the ROC curves with optimal probability thresholds and the classifier accuracy.

Model	AUC	Cut-off Value (as Probability)	Classifier Accuracy (%)	*p* Level
NLR + IL-1β	0.907	0.473	85.6	<0.001
NLR + IL-1Ra	0.841	0.405	83.3	<0.001
NLR + IL-6	0.908	0.544	87.8	<0.001
NLR + IL-8	0.896	0.458	88.9	<0.001
NLR + IL-10	0.887	0.520	84.4	<0.001
NLR + IL-13	0.830	0.441	83.3	<0.001
NLR + TNFα	0.853	0.501	82.2	<0.001
NLR + IFNγ	0.866	0.407	84.4	<0.001
NLR + HMGB1	0.823	0.449	83.1	<0.001

Abbreviations: AUC area under the ROC curve, NLR neutrophil/lymphocyte ratio, IL-1β interleukin 1β, IL-1Ra interleukin-1 receptor antagonist, IL-6 interleukin 6, IL-8 interleukin 8, IL-10 interleukin 10, IL-13 interleukin 13, TNFα tumor necrosis factor α, IFNγ interferon γ, HMGB1 high mobility group box 1.

**Table 5 ijms-24-16206-t005:** The characteristics of the ROC curves with optimal probability thresholds and the classifier accuracy.

Model	AUC	Cut-off Value (as Probability)	Classifier Accuracy (%)	*p* Level
SII + IL-1β	0.880	0.544	81.1	<0.001
SII + IL-1Ra	0.808	0.469	78.9	<0.001
SII + IL-6	0.872	0.459	81.1	<0.001
SII + IL-8	0.882	0.434	85.6	<0.001
SII + IL-10	0.845	0.444	76.7	<0.001
SII + IL-13	0.793	0.505	77.8	<0.001
SII + TNFα	0.816	0.542	77.8	<0.001
SII + IFNγ	0.794	0.513	77.8	<0.001
SII + HMGB1	0.783	0.445	78.7	<0.001

Abbreviations: AUC area under the ROC curve, SII systemic immune inflammation index, IL-1β interleukin 1β, IL-1Ra interleukin-1 receptor antagonist, IL-6 interleukin 6, IL-8 interleukin 8, IL-10 interleukin 10, IL-13 interleukin 13, TNFα tumor necrosis factor α, IFNγ interferon γ, HMGB1 high mobility group box 1.

**Table 6 ijms-24-16206-t006:** The clinical characteristics of patients with glial tumors.

		Value
Follow-up period	Mean ± SD (day) Median (range)	201 ± 152 174 (4–727)
Age at operation	Mean ± SD (year) Median (range)	66.0 ± 10.56 65.0 (38.4–88.9)
Gender	Females Males	30 (60%) 20 (40%)
Hemisphere	Left Right Midline or bilateral	24 (48%) 22 (44%) 4 (8%)
Location	Frontal lobe Temporal lobe Parietal lobe Occipital lobe Subtentorial location Multifocal	16 (32%) 15 (30%) 12 (24%) 4 (8%) 0 (0%) 3 (6%)
Ki-67	≥30% <30%	31 (62%) 19 (38%)
CRP	Mean ± SD (mg/L) Median (range)	10.59 ± 5.98 11.08 (0.85–21.99)

Abbreviations: Ki-67, a nuclear protein and a key marker associated with proliferating cancer cells, CRP C-reactive protein.

## Data Availability

The raw data supporting the conclusions of this article will be made available by the authors without undue reservation.

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
