# Peer review of "Cytokine Profile in Development of Glioblastoma in Relation to Healthy Individuals"

_ijms, 2023, doi:10.3390/ijms242216206_

Round 1
Reviewer 1 Report
Comments and Suggestions for Authors
In my opinion, the article is acceptable only if significant additional statistical analyzes and biochemical determinations are performed.
On what basis was the selection of the determined interleukins made? Is it only based on a literature review? Have any Luminex or Cytokine Array Kit screening tests been performed? If so, it would be worth including those results.
Additionally, a valuable complement to the work would be the determination of the CRP level.
It seems insufficient to compare only patients with GBM and healthy people. I believe that additional analyzes should be performed comparing the relationship between cytokine levels and clinical parameters of the tumor in a group of sick patients.
Reviewer 2 Report
Comments and Suggestions for Authors
This manuscript studied cytokine changes in glioblastoma patients and authors aimed to use these changes as molecular signatures for the glioblastoma patients. To achieve such aim, authors compared the difference, between patients and healthy individuals, of their white blood cells, immune cells, and cytokines. Although interesting, this manuscript needs to be polished in some places before publication. First, the abstract is not well written and contains too many abbreviations without clear definitions, leaving lots of confusion to our readers. The undefined terms include the rs (line 21), the AUC (line 22), the RR (line 22), the CI (line 22), and the ROC (line 25). Second, this study lacks enough experimental support to show these cytokine changes happen exclusively in the glioblastoma disease. I did not see evidence to suggest that these changes do not show in other diseases including other cancers. In other words, if other diseases also have such changes, we might not be able to use them as molecular signatures for the glioblastoma. Third, although the discussion are clearly written, it is more like a literature review instead of data interpretation in a research article. Authors should give more insights in the discussion and heavily cut off irrelevant parts of “literature review”. Overall, this manuscript interestingly shows some clinical analysis of glioblastoma patients and thus can provides some good data reference for the field of glioblastoma studies.
Comments on the Quality of English LanguageEnglish is fine.
Round 2
Reviewer 1 Report
Comments and Suggestions for Authors
Dear Authors, although there is no point in performing screening tests at this stage of the research, I regret that they were not performed earlier. Although you are testing quite a large number of proteins, next time I suggest performing broad-spectrum array tests before using ELISA tests for single proteins.
It is obvious to me that CRP is not a specific biomarker, but I believe that measuring this parameter is advisable if you are testing for pro-inflammatory factors. I'm glad this has been completed.
Although I still believe that the research is at an early stage and interleukins will probably not prove to be specific biomarkers in glioma, I believe that the work submitted to me for review is interesting and worth publishing.